# Profile of MicroRNAs Associated with Death Due to Disease Progression in Metastatic Papillary Thyroid Carcinoma Patients

**DOI:** 10.3390/cancers15030869

**Published:** 2023-01-31

**Authors:** Ana Kober Leite, Kelly Cristina Saito, Thérèse Rachell Theodoro, Fátima Solange Pasini, Luana Perrone Camilo, Carlos Augusto Rossetti, Beatriz Godoi Cavalheiro, Venâncio Avancini Ferreira Alves, Luiz Paulo Kowalski, Maria Aparecida Silva Pinhal, Edna Teruko Kimura, Leandro Luongo Matos

**Affiliations:** 1Head and Neck Surgery Department, Instituto do Câncer do Estado de São Paulo, Hospital das Clínicas da Faculdade de Medicina da Universidade de São Paulo, São Paulo 05403-010, SP, Brazil; 2Faculdade Israelita de Ciências da Saúde Albert Einstein, Hospital Albert Einstein, São Paulo 05653-120, SP, Brazil; 3Department of Cell and Developmental Biology, Institute of Biomedical Sciences, University of São Paulo, São Paulo 05508-000, SP, Brazil; 4Discipline of Biochemistry, Department of Morfology and Physiology, Faculdade de Medicina Do ABC, São Paulo 09060-870, SP, Brazil; 5Centro de Investigação Translacional em Oncologia, Instituto do Câncer do Estado de São Paulo Paulo (ICESP), Hospital das Clínicas (HCFMUSP), Faculdade de Medicina da Universidade de São Paulo, Sao Paulo 01246-000, SP, Brazil; 6Pathology Department, Instituto do Câncer do Estado de São Paulo, Laboratório de Investigação Médica 14 (LIM14), Faculdade de Medicina da Universidade de São Paulo, São Paulo 01246-903, SP, Brazil

**Keywords:** papillary thyroid carcinoma, prognosis, death, molecular biology, microRNAs

## Abstract

**Simple Summary:**

Papillary thyroid cancer is not an aggressive cancer, even when metastasis are present; therefore, treatment has been downgraded in recent years. However rare, mortality exists and finding the factors associated with mortality is essential and has not yet been fully accomplished; the answer probably lies in molecular factors. This study aimed to find MicroRNAs related to death in metastatic patients and found that patients who died due to progression of PTC had higher expression levels of miR-101-3p, miR-17-5p, and miR-191-5p compared with those of patients with stable metastatic disease. These findings are new in the literature and may open new doors in differentiating the patients who need more aggressive treatment from those who do not.

**Abstract:**

Papillary thyroid carcinoma (PTC) is the most common neoplasm of the endocrine system and has an excellent long-term prognosis, with low rates of distant metastatic disease. Although infrequent, there are cases of deaths directly related to PTC, especially in patients with metastatic disease, and the factors that could be associated with this unfavorable outcome remain a major challenge in clinical practice. Recently, research into genetic factors associated with PTC has gained ground, especially mutations in the TERT promoter and BRAF gene. However, the role of microRNAs remains poorly studied, especially in those patients who have an unfavorable outcome at follow-up. This paper aims to evaluate molecular markers related to the different pathological processes of PTC, as well as the histological characteristics of the neoplasm, and to compare this profile with prognosis and death from the disease using an analysis of patients treated for metastatic disease in a single tertiary cancer center. Evaluation of microRNA expression in paraffin-embedded tumor specimens was carried out by quantitative PCR using the TaqMan^®^ Low Density Array (TLDA) system. Metastatic patients who died from progression of PTC had higher expressions of miR-101-3p, miR-17-5p, and miR-191-5p when compared to patients with stable metastatic disease. These findings are of great importance but should be considered as preliminary because of the small sample.

## 1. Introduction

Papillary thyroid carcinoma (PTC) is the most common malignant neoplasm of the thyroid gland; however, it represents only 1% of all diagnosed cases of cancer and approximately 0.2% of deaths from this disease [1,2]. In most patients, it is curable and has an indolent course and excellent prognosis. The literature shows a 10-year disease-specific mortality of less than 5%. However, despite the low mortality, there are cases of deaths directly related to PTC, and factors that may be associated with unfavorable outcomes in these cases remain a major challenge in clinical practice [3].

In recent years, a series of genetic and molecular alterations related to the genesis of PTC have been studied and partially defined [4]. It is known that the initiation and progression of PTC involve the accumulation of several genetic and epigenetic modifications [5,6].

MicroRNAs (miRNAs) are small endogenous non-coding RNAs that mediate the regulation of gene expression at the post-transcriptional level. The expression of miRNAs plays an important role in various physiological processes, including cell cycle control, differentiation, proliferation, apoptosis, cell homeostasis, and organogenesis. The dysregulation of miRNA expression is considered as an important step in tumor development and progression in several human tissues. The overexpression of specific miRNAs can lead to the suppression of tumor suppressor genes, and the underexpression of other miRNAs can result in the increased expression of oncogenes; both situations affect cell proliferation, differentiation, and apoptosis, leading to tumor growth and progression. miRNAs have also been associated with several stages of the invasion–metastasis cascade in cancer, such as modification of the tumor microenvironment, local invasion, survival in the vasculature, and proliferation at distant sites [7]. However, the exact biological role of miRNAs in thyroid carcinogenesis remains uncertain. In an independent study, researchers analyzed the expression of miRNAs in different thyroid tumors and showed the dysregulation of miRNAs in tumor tissues compared with normal ones [8].

In recent decades, there has been a change in the causes of death related to PTC. Early series reported mortality as mainly related to local disease progression, whereas recent series showed distant metastasis (DM) as the main cause of death, which may be explained by the increase in aggressive resections and widespread application of radioactive iodine (RAI) therapy [1]. However, why histologically similar tumors have an indolent course in some cases but lead to tumor progression and death in other cases remains unknown. This is also observed among metastatic PTC patients; some patients have stable and asymptomatic disease for many years, whereas others have aggressive behavior, leading to death. Therefore, the present study evaluated the expression of miRNAs related to death in patients with metastatic PTC.

## 2. Materials and Methods

### 2.1. Data Survey

Among all patients with PTC treated by the Head and Neck Surgery Department at the Instituto do Câncer do Estado de São Paulo, Hospital das Clínicas of the Faculdade de Medicina, Universidade de São Paulo (Icesp, HCFMUSP), 108 patients were identified as having DM (at the time of initial diagnosis of PTC or during follow-up). These patients were fully managed at the Institution or referred from another location after the initial surgery. Initial diagnosis and surgical treatment were performed from 1986 to 2015 and, during this period, 3555 patients were surgically treated for PTC at the institution.

The pathological stage system (pTNM) was revised according to the 8th edition of the AJCC/UICC Cancer Staging Manual [9] and initial stage (at the time of diagnosis) was used.

The diagnosis of DM was based on clinical and/or positive findings on iodine-131 whole-body scanning (WBS), radiography (chest X-ray), computed tomography (CT), MRI, or tissue biopsy.

Dedifferentiation was defined based on the pathological examination of the tumor or metastatic disease and considered when there was an abrupt transformation from a well-differentiated tumor to a tumor with a high-grade morphology without the original distinctive histological features [10].

Iodine avidity was determined by visual uptake at the known site of metastatic disease on WBS after the first adjuvant treatment with RAI under hypothyroid conditions. The tumor was considered refractory to RAI therapy based on the 2015 American Thyroid Association Guidelines [11].

Patients were considered to have died from PTC if extensive, symptomatic, and progressive metastatic disease was present at the last follow-up and it was declared as the primary cause of death or as a significant contributing cause of death on the death certificate.

### 2.2. Selection of Samples for Molecular Study

The material used in the molecular analyses was a small part of the tumor remnant in formalin-fixed paraffin-embedded (FFPE) blocks obtained from anatomopathological examination.

For molecular analyses, all patients who died from the disease and had representative tumor material from FFPE samples at the institution were included, and among these patients, paired samples were selected at a 2:1 ratio from a second group of metastatic patients with a long follow-up period. Therefore, 16 patients who died due to PTC progression and 8 “controls” (i.e., metastatic patients with stable disease during follow-up) were included, giving a total of 24 patients (samples). The final sample consisted of 17 deceased patients and 7 controls (one patient died of DM during follow-up and at the end of the analyses).

### 2.3. Sequencing for BRAF and TERT Mutations

For *TERT* promoter and *BRAF* mutation analyses genomic DNA extraction was performed using a nucleic acid isolation protocol with FFPE samples as described by Kizys et al. [12]. The amplification of the TERT promoter and BRAF sequences was performed by nested PCR. In brief, an extended sequence of the gene was amplified in the first PCR (PCR I), and the product was used as a template in the second round of PCR (PCR II). The specific primer pairs for each sequence were designed by the Primer-BLAST program [13] and are presented in Appendix A. The PCR reaction mixture in a final volume of 25 µL was prepared with 400 ng of genomic DNA, 400 nM primers, 5 µL of MyTaq DNA Polymerase Buffer 5×, 0.2 mM dNTP, and 1 µL of MyTaq DNA Polymerase (Bioline^®^; Meridian Bioscience^®^, Cincinnati, OH, USA). The amplification was performed using the 2720 Thermal Cycler (Applied Biosystems, Thermo Fischer Scientific^®^, Waltham, MA, USA) under the following conditions: pre-denaturation at 95 °C for 5 min, followed by 40 cycles of denaturation at 94 °C for 30 s, annealing at 51–60 °C for 35–40 s, and extension at 72 °C for 35 s, and a final extension phase at 72 °C for 7 min. Therefore, 2 µL of the amplified product from PCR I was used as the template for PCR II. The final product was recovered from 1% agarose gel electrophoresis and purified using the QIAquick Gel Extraction Kit (Qiagen^®^, Hilden, Germany). For capillary sequencing by the Sanger method, the sample was mixed with a fluorescent nucleotide solution according to the manufacturer’s recommendations (BigDye^®^ Terminator v3.1 Cycle Sequencing Kit; Applied Biosystems, Thermo Fischer Scientific^®^, Waltham, MA, USA) and amplified using the ABI 3730 DNA Analyzer (Applied Biosystems, Thermo Fischer Scientific^®^, Waltham, MA, USA). The generated sequences were analyzed by Sequence Scanner Software v2.0 (Applied Biosystems, Thermo Fischer Scientific^®^, Waltham, MA, USA), and the identification of mutational points was obtained using an alignment program in the BLAT database [14].

### 2.4. MicroRNA Detection Technique

Sixty-four microRNAs were selected for this study. The selection was carried out based on an extensive literature review, including miRNAs previously associated with PTC, tumor aggressiveness, metastatic disease, and invasion processes such as epithelial mesenchymal transition in thyroid cancer and other tumors.

The level of miRNA expression was assessed by reverse transcription-quantitative polymerase chain reaction (RT-qPCR) according to a previously stablished procedure [15]. RNA was extracted from 10 sections (5 μm) of paraffin-embedded tissue using the MagMAX FFPE RNA Ultra Kit (Applied Biosystems, Foster City, CA, USA) according to the manufacturer’s protocol. RT-qPCR was performed using the TaqMan Low Density Array system (Thermo Fisher Scientific^®^, Waltham, MA, USA). The RNA samples were subjected to reverse transcriptase assay with polyadenylation to obtain the cDNA product. All amplifications were performed in triplicate, and the cycle threshold (CT) values were determined with a cloud platform (Thermo Fisher Scientific^®^, Waltham, MA, USA). For methodological rigor, miRNAs with CT values > 36 were regarded as indeterminate and not included. In addition, only miRNAs detected in at least 70% of the samples were considered for statistical analysis. Therefore, the following 18 miRNAs were not included in the analyses: miR-129-5p, miR-130b-3p, miR-137-3p, miR-138-2-3p, miR-146b-3p, miR-155-3p, miR-17-3p, miR-187-3p, miR-302c-3p, miR-30e-5p, miR-34b-3p, miR-34c-5p, miR-455-3p, miR-4788, miR-506-3p, miR-654-3p, miR-9-5p, and miR-98-5p. miRNA expression was normalized between samples using the quantile method with the Expander software [16] selecting the most stable expressions among the samples (endogenous normalizers). The mean of the miRNAs let-7g-5p and miR-181a-5p was used to normalize the expression values of the remaining miRNAs. The expression levels of these miRNAs were quantified using the 2-ΔCt method to identify upregulated or downregulated miRNAs.

Commercially available primers sold by Thermo Fisher Scientific^®^ were used and are listed in Appendix A. They are constructed based on data available at miRbase.org.

### 2.5. Statistical Analyses

For the statistical analyses, the values of continuous variables are presented as the mean and standard deviation (SD) or standard error (SE). For categorical variables, absolute and relative frequencies are presented. The cutoff values of quantitative variables were determined by receiver operating characteristic (ROC) analysis, and the cutoff values for qualitative variables were determined based on the association of categories. For the comparison of two sample populations, the Mann–Whitney test was used, and for three or more populations, the Kruskal–Wallis test with Dunn’s auxiliary test was employed. Comparisons of the frequency of a phenomenon between groups for categorical variables were performed with the chi-square test or Fisher’s exact test. Variables with *p* < 0.10 in univariate analysis were used for multivariate analysis, which was performed using linear regression models for independent quantitative variables. The inclusion of variables with *p* < 0.10 in the multivariate analysis excludes possible dependent and confounding variables. In addition to sensitivity and specificity, accuracy estimates (area under the ROC curve—AUC) were determined by ROC curve analysis (positive Z-scores were used in cases of upregulated miRNAs, and negative Z-scores were used in the cases of downregulated miRNAs). In all analyses, SPSS version 28.0 (IBM^®^, Endicott, NY, USA) was used, and the Multiple Experiment Viewer (MEV version 4.5.0; available at: http://mev.tm4.org, accessed on 22 February 2022) was used to obtain the heatmaps.

## 3. Results

The descriptive data of the patients included in the study are shown in Table 1. The group of patients who died from disease progression had more metastatic disease at multiple sites and demonstrated disease dedifferentiation, and there was a higher number of RAI-refractory patients. For other variables, the distribution was relatively homogeneous between the two groups, including age, stage, histopathological characteristics, and treatment received. Notably, we selected patients with a long follow-up period among metastatic patients with stable disease as we aimed to exclude follow-up time as a possible bias. Therefore, the follow-up time was approximately three times longer in the group of patients alive at the end of the follow-up compared with that of patients who died due to disease progression.

Four patients (23.5%) who died from the disease had a mutation in the BRAF gene, and eight patients (47%) had a mutation in the TERT promoter gene. Three individuals had both mutations. In the group of metastatic patients with stable disease, the rates of mutations in the BRAF and TERT genes were 28.6% (two cases) and 100%, respectively. In addition, BRAF and TERT mutations were present in 25% and 62.5% of the sample of metastatic individuals, respectively, regardless of the outcome (death or alive with stable disease).

The let-7c-5p, let-7e-5p, 101-3p, 16-5p, 17-5p, 181b-5p, 191-5p, 19a-3p, 19b-3p, 200c-3p, 20a- 5p, 29a-3p, 30b-5p, 30c-5p, 30d-5p, 31-5p, and 34a-5p miRNAs were differentially expressed between the two groups in univariate analysis, as shown in Table 2.

miRNAs with *p*-values lower than 0.1 were selected for multivariate linear regression analysis to identify those expressed independently. The data for the miRNAs that were significant in the multivariate analyses are shown in Table 3 and the variables that were included but not significant are shown in Appendix A. In comparison with patients with stable metastatic disease, patients who died due to disease progression had higher expression levels of miR-101-3p, miR-17-5p, and miR-191-5p. Three models based on miR-17-5p alone; miR-17-5p in association with mir-101-3p; and miR-101-3p, miR-17-5p, and miR-191-5p together were proposed, as shown in Table 3.

The results of heatmap analysis using the three proposed models with the miRNAs differentially expressed in multivariate analysis (overexpressed miR-17-5p, miR-101-3p, and miR-191-5p) are presented in Figure 1. A clear segregation between patients who died from PTC progression compared with metastatic patients without disease progression was achieved, especially when considering the association of the three selected miRNAs (model 3).

Values greater than 0.861, 2.155, and 0.455 were established as the ROC curve cutoff values of miR-101-3p, miR-17-5p, and miR-191-5p, respectively, for the evaluation of death of patients with metastatic PTC. The three models showed high accuracy in predicting the mortality of patients with metastatic PTC, as shown in Table 4. None of the patients with these microRNAs expression bellow the cutoff value died; therefore, the specificity is 100%. However, the low number of patients in both groups did not allow the analysis of the risk of death by logistic regression of the cutoff values of the three miRNAs.

## 4. Discussion

The present study demonstrated the overexpression of the miRNAs 17-5p, 101-3p, and 191-5p in patients who died due to metastatic PTC progression. Furthermore, the expression of the three miRNAs was lower in metastatic patients with an indolent course (defined as alive and without metastatic disease progression in a long follow-up period) than in patients with worse PTC progression. This is the first study to identify a different profile of miRNAs associated with death in PTC patients, the knowledge of which could facilitate the understanding of the biology of these tumors and contribute to accurate follow-up and specific therapeutics in the future.

Various miRNAs in PTC samples were previously evaluated in different studies in the last decade. However, most studies compare PTC with normal tissue while the present study included only metastatic PTC patients. Liu et al. [17] evaluated a large profile of miRNAs in 126 samples from PTC patients by comparing their expression levels with those in matching normal thyroid tissues in order to understand the role of miRNAs in PTC carcinogenesis. The authors identified the overexpression and underexpression of 180 and 68 miRNAs in cancer samples, respectively, including the overexpression of miR-222 (target genes KIT and AXIN2), miR-15a (target genes AXIN2 and FOXO1), and miR-221 (target gene KIT) and the underexpression of miR-206 (target gene MET), miR-299-3p (target gene ITGAV), miR-103 (target gene ITGA2), and miR-101 (target gene ITGA3), which was also downregulated in the group of patients with indolent metastatic disease in our study. In another study, Rosignolo et al. [18] evaluated the expression of 754 different miRNAs in PTC patients. The authors identified eight overexpressed miRNAs (miR-221-3p, miR-222-3p, miR-146a-5p, miR-24-3p, miR-146b-5p, miR-103a-3p, miR-28-3p, and miR-191-5p) in serum of PTC patients compared with patients with benign nodules. Similarly, miR-191-5p was overexpressed in the group of patients who died due to PTC progression in our study. Serum levels of miR-146a-5p and miR-221-3p were also associated with worse response to therapy during follow-up. Bioinformatic analysis for microarray and RNA-seq has also been used for identification of potential target miRNAs. Using these methods, Wang et al. [19] identified miR-146b-5p, miR-15a-5p, miR-21-5p, miR221-3p, and miR-222-3p as differently expressed in PTC compared to normal tissue.

There are limited studies on the three differentially expressed miRNAs identified in our study. Moreover, the evaluation of death as an outcome for metastatic PTC patients is unique in the literature, thus limiting the comparison of our findings with those of previous studies.

miR-17-5p is a member of the miR-17-92 cluster, located on chromosome 13q31-32. It was previously described as related to tumor proliferation, invasion, migration, and apoptosis inhibition in thyroid cancer, targeting PTEN and other pathways [20,21,22]. Specifically, miR-17 has been identified as upregulated in anaplastic thyroid carcinoma (ATC) compared with adjacent normal thyroid tissues [23,24]. In another study, the inhibition of the expression of this miRNA was associated with growth arrest and the induction of anaplastic cell apoptosis via caspase activation [25]. However, another group found that the inhibition of miR-17 increased MYCN and c-MYC expression, which increased pri-miR-17-92 transcripts and the expression of oncogene miRNAs (miR-18a and miR-19a), demonstrating that miR-17 inhibition did not regulate thyroid tumor growth in two different cell lines of anaplastic thyroid carcinoma [26]. The conflicting results suggest that different miRNAs have different expression levels and functions depending on the tumor microenvironment and behavior. This notion is consistent with the findings of Liu et al., who observed that miR-17-5p exhibited a dual function in tumor progression [27]. Additionally, the increased serum level of miR-17-5p was found in aggressive cases of medullary thyroid cancer compared with PTC [28].

miR-101 is an important tumor suppressor and its underexpression is associated with thyroid malignancy. It is also considered as an “anti-metastatic” miRNA in PTC, which can inhibit migration by targeting the Rac1 gene [29,30,31] and induce apoptosis by enhancing the TRAIL pathway [32]. In previous studies, miR-101-3p overexpression inhibited the invasion and migration of PTC cells in vitro through different pathways [33,34]. Zhao et al. [35] demonstrated that miR-101 reduced PTC cell proliferation, apoptosis resistance, and invasion by targeting USP22. In addition, its underexpression was associated with lymph node metastasis and poor prognosis (lower survival) among PTC patients, which is consistent with our results showing the underexpression of this miRNA in metastatic PTC patients. Nevertheless, we also observed the higher expression of miR-101 in metastatic cases that progressed to death, demonstrating that the precise mechanism of miR-101-3p in aggressive PTC remains unclear.

The miR-101 gene has two copies, hsa-miR-101.1 located in the intron of the LOC107983962 gene (hg38: chr1:65,058,434–6,505,850 [−]) and hsa-miR101.2 located in the intron of the RCL1 gene (hg38: chr9:4,850,297–4,850,375 [+]). Furthermore, the miRNA is not a single sequence but a series of multiple isomiRs with expression and sequence diversity. Guo et al. [36] estimated five isomiRs with two seeds from the miR-101-3p.2 locus, showing a significant difference in gaining or losing target mRNAs. Deregulated isomiRs may also contribute to the abnormal expression patterns of their target mRNAs, and lncRNAs may disrupt the isomiR:mRNA network, further controlling mRNA expression.

In some tumors, miR-101-3p was confirmed to target the Moloney murine leukemia virus 1 (Pim-1) oncogene [37]. The oncogenic Pim kinase proteins (Pim-1/2/3) regulate tumorigenesis through the phosphorylation of essential proteins that control the cell cycle and proliferation [38,39,40]. Therefore, miR-101-3p could be considered as a potential antineoplastic molecule. However, unlike most kinases, Pim does not have regulatory domains; once translated, it is constitutively active [41]. Taken together, the findings suggest that despite miR-101-3p promoting a decrease in the synthesis of Pim kinases, the level of these proteins may not be affected as their degradation could be inhibited. Therefore, even with the overexpression of miR-101-3p, the level of Pim kinases might be high, which would explain the worse prognosis of patients with PTC [42,43]. In addition, researchers have recently demonstrated the role of Pim kinases in immunomodulation; their increase may cause tumor immune escape and help tumor cells survive [44].

MDM4 directly binds to p53, resulting in degradation. Therefore, it is considered as a negative regulator of the p53 suppression pathway and has been identified as a target site for miR-191 [45]. The binding of miR-191 to MDM4-C allele mRNA, which could result in the increased expression of MDM4, may explain tumor progression in different types of cancers such as ovarian cancer and retinoblastoma [46,47]. The overexpression of miR-191-5p was reported only in the serum of PTC patients compared with normal thyroid tissues [18], indicating its role in thyroid carcinogenesis. Furthermore, Colamaio et al. [48] reported the high expression of miR-191 in PTC and ATC and its downregulation in follicular thyroid carcinoma and the follicular variant of PTC, suggesting the role of this miRNA in thyroid carcinogenesis. This is the first report of the association of miR-191-5p with the aggressive behavior of PTC.

The strength of the present study is that it is the first to evaluate the role of different miRNAs in death due to tumor progression in metastatic PTC patients, a rare but fatal event. The rarity of this event makes it difficult to study in PTC, and large series are not available. Nevertheless, the small sample size is a limitation of our study, and our results should be considered as preliminary. However, since commercially available primers were used, the results can be reproduced. 

Some discrepancies between the overexpression of the three miRNAs identified in the present study and their underexpression in cases of poor treatment response or outcome reported in the literature, mostly in in vitro studies, may be attributed to a possible type I error; however, this is a statistically remote possibility. A more plausible interpretation of the findings would be that the role of the differentially expressed miRNAs has not been well established for this particular outcome. Another limitation is the fact that this study only analyzed a selected group of microRNAs; therefore, there could be other miRNAs significant to this outcome that where not evaluated.

## 5. Conclusions

The present study demonstrated that metastatic patients who died due to the progression of PTC had higher expression levels of miR-101-3p, miR-17-5p, and miR-191-5p compared with those of patients who had stable metastatic disease. Further studies on the miRNA profile of patients with unfavorable outcomes are required to better understand the role of miRNAs not only in thyroid carcinogenesis but also in the development of metastatic disease, tumor progression, and death.

## Figures and Tables

**Figure 1 cancers-15-00869-f001:**
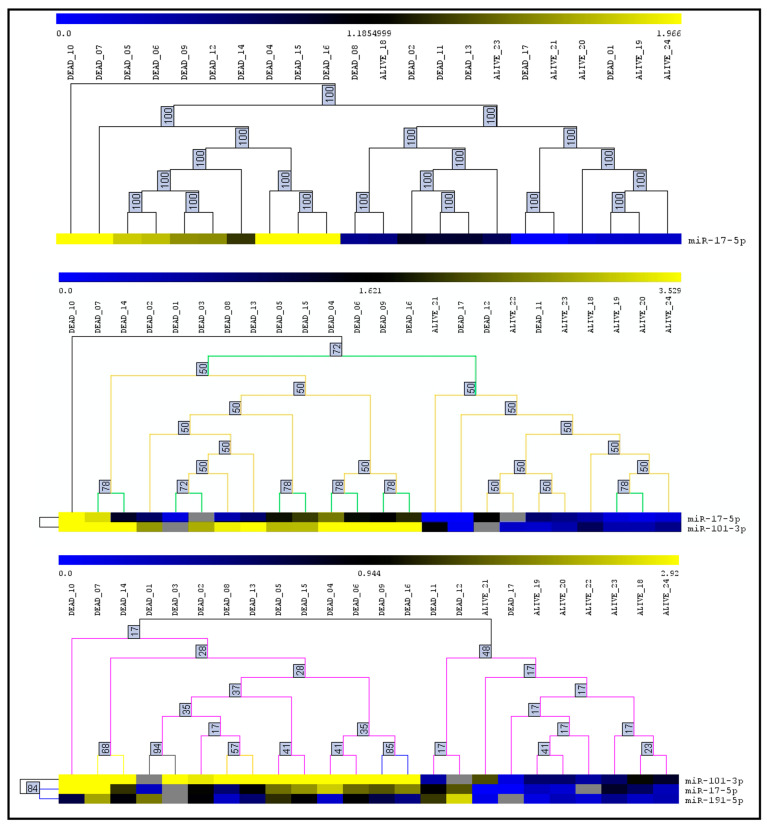
Heatmaps demonstrating the differently expressed microRNAs and hierarchical cluster analyses of all samples at the differentiation between dead patients due to papillary thyroid carcinoma progression from those alive with stable metastatic disease (without tumor progression at long follow-up), considering only the microRNAs differently expressed at the multivariate analysis (overexpression of miR-17-5p, miR-101-3p, and miR-191-5p). Yellow spots represent samples with high expression and those in blue low represent microRNAs expression.

**Table 1 cancers-15-00869-t001:** Descriptive data of both groups of patients with metastatic papillary thyroid carcinoma included in the study.

Variable	Death Due to Cancer Progression n = 17 (%)	Alive Metastatic Patients n = 7 (%)
**Sex**		
Female	12 (70.6)	4 (57.1)
Male	5 (29.4)	3 (42.9)
**Age**		
Mean ± SD (years-old)	54.8 ± 13.2	56 ± 13.4
**Histopathological data**		
Extrathyroidal extension	9 (52.9)	3 (50.0)
Vascular invasion	11 (73.3)	4 (66.7)
Multifocality	9 (52.9)	4 (57.1)
Size (mean ± SD; cm)	4.7 ± 3.6	4.1 ± 2.6
**Initial pT classification**		
pT1	4 (23.5)	1 (16.7)
pT2	2 (11.8)	1 (16.7)
pT3	5 (29.4)	2 (33.3)
pT4a	2 (11.8)	2 (33.3)
pT4b	4 (23.5)	0
**Initial pN classification**		
pN0	8 (50)	2 (28.6)
pN1a	3 (18.8)	1 (14.3)
pN1b	5 (31.3)	4 (57.1)
**Treatment**		
Total thyroidectomy	16 (94.1)	7 (100)
Central compartment neck dissection	8 (50)	5 (71.4)
Level II-V neck dissection	5 (31.3)	4 (57.1)
R0 surgery	13 (86.7)	6 (85.7)
Radioiodine therapy (RIT)	12 (70.6)	7 (100)
**Number of RIT**		
One	6 (50)	4 (57.1)
>2	6 (50)	3 (42.9)
**Distant metastases**	17 (100)	7 (100)
Lung	16 (94.1)	7 (100)
Liver	4 (23.5)	0
Bones	12 (70.6)	1 (14.3)
Multiple sites	14 (82.4)	1 (14.3)
Diagnosis with the primary tumor	12 (70.6)	4 (57.1)
Dedifferentiation	8 (47.1)	0
**Follow-up**		
Radioiodine refractory disease	9 (75)	3 (42.9)
Regional failure	7 (50)	1 (14.3)
Follow-up time (mean ± SD; months)	51.9 ± 37.9	158.6 ± 32.9

**Table 2 cancers-15-00869-t002:** MicroRNAs differently expressed between death patients due to progression of metastatic papillary thyroid carcinoma and those alive with stable metastatic disease (without tumor progression at long follow-up).

miR	Death Due to Tumor Progression		Alive with Metastatic Disease		*p*-Value (Mann–Whitney)
	MEAN	SE	MEAN	SE	
let-7b-5p	0.082	0.036	0.076	0.038	0.413
**let-7c-5p**	**5.185**	**3.002**	**5.841**	**1.105**	**0.020**
let-7d-5p	0.445	0.052	0.470	0.064	0.619
**let-7e-5p**	**2.981**	**1.243**	**5.490**	**1.670**	**0.020**
let-7f-5p	3.855	0.748	4.950	1.164	0.494
let-7i-5p	20.463	3.414	11.609	1.268	0.089
miR-1-3p	17.129	7.263	14.596	6.562	0.891
**miR-101-3p**	**3.863**	**0.517**	**0.776**	**0.154**	**0.005**
miR-10b-5p	0.471	0.124	0.774	0.303	0.590
miR-125a-5p	7.230	1.550	7.422	1.633	0.664
miR-138-5p	3.919	0.906	1.811	0.618	0.081
miR-141-3p	17.303	3.336	9.412	2.434	0.172
miR-146b-5p	2.686	0.913	4.957	1.419	0.179
**miR-16-5p**	**5.610**	**1.402**	**0.926**	**0.144**	**0.019**
**miR-17-5p**	**1.811**	**0.417**	**0.351**	**0.117**	**0.005**
**miR-181b-5p**	**0.262**	**0.048**	**0.474**	**0.053**	**0.014**
miR-18a-5p	0.058	0.025	0.096	0.094	0.364
**miR-191-5p**	**1.057**	**0.181**	**0.236**	**0.044**	**<0.001**
miR-199a-3p	1.328	0.512	0.752	0.230	0.757
**miR-19a-3p**	**0.566**	**0.212**	**0.053**	**0.019**	**0.003**
**miR-19b-3p**	**2.220**	**1.134**	**0.125**	**0.035**	**<0.001**
miR-200a-3p	3.842	0.642	2.183	0.386	0.209
miR-200b-3p	38.351	11.600	51.367	9.258	0.065
**miR-200c-3p**	**16.783**	**4.204**	**33.108**	**4.673**	**0.011**
miR-203a-3p	0.101	0.044	0.056	0.032	0.773
miR-205-5p	2.791	1.154	1.203	0.691	0.602
**miR-20a-5p**	**0.526**	**0.111**	**0.124**	**0.033**	**<0.001**
miR-21-5p	55.030	19.916	19.751	5.168	0.383
miR-214-3p	0.965	0.503	0.094	0.062	0.100
miR-221-3p	23.834	5.939	18.518	5.118	1.000
miR-222-3p	1.667	0.406	1.721	0.312	0.452
**miR-29a-3p**	**6.587**	**1.112**	**3.260**	**0.399**	**0.006**
miR-30a-5p	5.969	1.762	2.299	0.433	0.072
**miR-30b-5p**	**15.418**	**3.490**	**4.958**	**0.964**	**0.018**
**miR-30c-5p**	**26.561**	**5.902**	**3.662**	**0.694**	**0.005**
**miR-30d-5p**	**1.930**	**0.383**	**0.663**	**0.171**	**0.021**
miR-30e-3p	0.691	0.154	0.550	0.201	0.559
**miR-31-5p**	**2.252**	**1.922**	**1.403**	**0.371**	**0.017**
**miR-34a-5p**	**3.751**	**1.037**	**5.668**	**0.950**	**0.032**
miR-423-5p	0.454	0.093	0.896	0.267	0.091
miR-429	0.366	0.097	0.147	0.053	0.178
miR-483-3p	1.298	0.752	0.289	0.103	0.831
miR-92a-3p	15.574	7.610	4.612	0.887	0.671

Legend: SE = standard-error. Highlighted in bold are the variables with *p* < 0.05.

**Table 3 cancers-15-00869-t003:** Multivariate analyses (linear regression) identifying microRNAs of risk of death due to tumor progression in patients with metastatic papillary thyroid carcinoma.

Model		Unstandardized Coefficients		Standardized Coefficients	*p*-Value	95% CI (for Beta)	
		B	IF	Beta		Lower	Upper
1	Constant	−0.021	0.113		0.860	−0.310	0.268
	**miR-17-5p**	**0.555**	**0.073**	**0.959**	**0.001**	**0.367**	**0.743**
2	Constant	−0.122	0.079		0.199	−0.343	0.098
	**miR-17-5p**	**0.451**	**0.059**	**0.779**	**0.002**	**0.288**	**0.614**
	**miR-101-3p**	**0.066**	**0.023**	**0.294**	**0.044**	**0.003**	**0.129**
3	Constant	−0.183	0.036		0.015	−0.298	−0.068
	**miR-17-5p**	**0.341**	**0.035**	**0.589**	**0.002**	**0.230**	**0.452**
	**miR-101-3p**	**0.071**	**0.010**	**0.318**	**0.005**	**0.041**	**0.102**
	**miR-191-5p**	**0.265**	**0.060**	**0.231**	**0.021**	**0.075**	**0.456**

Legend: SE = standard-error. Highlighted in bold are the variables with *p* < 0.05.

**Table 4 cancers-15-00869-t004:** Evaluation of microRNAs 17-5p, 101-3p, and 191-5p for death due to tumor progression prediction in patients with metastatic papillary thyroid carcinoma.

Micro-RNA	Sensitivity	Specificity	AUC	95% CI (AUC)	
				Lower	Upper
**Model 1** miR-17-5p ≥ 0.861	0.923	1.000	0.962	0.872	1.051
**Model 2** miR-17-5p ≥ 0.861 AND miR-101-3p ≥ 2.155	0.786	1.000	0.893	0.750	1.036
**Model 3** miR-17-5p ≥ 0.861 AND miR-101-3p ≥ 2.155 AND miR-191-5p ≥ 0.455	0.714	1.000	0.857	0.694	1.021

Legend: AUC = area under the ROC curve.

## Data Availability

The data presented in this study are available on request from the corresponding author.

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
