# Peer review of "Profile of MicroRNAs Associated with Death Due to Disease Progression in Metastatic Papillary Thyroid Carcinoma Patients"

_cancers, 2023, doi:10.3390/cancers15030869_

Round 1

Reviewer 1 Report

Leite et al work select miRNAs that are significantly based on a literature review to construct pcr primers to identify the expression level of those in papillary thyroid carcinoma samples. The goal was to identify a miRNA prognostic biomarker panel for the death of these patients. The authors identified miRNAS differentially expressed and tried to build a logistic regression model for that. The idea is interesting, however with the amount of public available high-throughput data in this moment, it is a little bit weird that the authors even didn't cite any work with miRNA sequencing in this type of cancer. The Cancer Genome Atlas has papillary thyroid carcinoma miRNA sequenced. Works such as Wang et al. 2020 (PMC6906535) use this data and they even cite hsa-miR-15a-5p as important which wasn't used in the current work.  So the major suggestion is to compare with high-throughput data , smallRNA-seq or miRNA-seq data , or even miRNA array data. Or the authors should make primers for all human miRNAs to prove that they are using the correct set.

Small comments :

line 220 : are you using 0.1 or 0.01 threshold? and in addition why you didn't move miR-29a-3p or miR-30c-5p or miR-20a-5p to the logistic regression, they also have significant p-values?

figure 1: I understood that the authors were building heatmap as they were building the logistic regression model, but it isn't the correct way to build a heatmap. I would recommend to build a heatmap with all p-value signifcant miRNAs and a second one with only the miRNAs selected in the final model.

Author Response

We greatly appreciate the excellent review and valuable comments made on our work. After careful revision, we identified some methodological steps that were not presented as clearly as intended and may have been the reason for some of the comments.

All corrections were made on the document and are listed below.

Once again, thank-you for your valuable comments and opportunity to review our work for your prestigious journal.

Comment: “Leite et al work select miRNAs that are significantly based on a literature review to construct pcr primers to identify the expression level of those in papillary thyroid carcinoma samples.”

Response: We did not construct the primers, we used commercially available primers sold by Thermo Fisher Scientific® that are based on the data available on miRbase.org.

An example for the Thermofisher Datasheet for miR-101-3p is attached. 

This information was not clear initially and was included in “Methods”

Comment: “The idea is interesting, however with the amount of public available high-throughput data in this moment, it is a little bit weird that the authors even didn't cite any work with miRNA sequencing in this type of cancer. The Cancer Genome Atlas has papillary thyroid carcinoma miRNA sequenced. Works such as Wang et al. 2020 (PMC6906535) use this data and they even cite hsa-miR-15a-5p as important which wasn't used in the current work.  So the major suggestion is to compare with high-throughput data , smallRNA-seq or miRNA-seq data , or even miRNA array data”

Response: The main papers based on large arrays of miRs in PTC patients are discussed in the second paragraph of Discussion and bioinformatic data was included.

Comment: “Or the authors should make primers for all human miRNAs to prove that they are using the correct set.”

Response: as explained above, we did not make the primers but used commercially available ones based on data from miRbase, an advantage being that therefore the results can be reproduced.

Comment: “are you using 0.1 or 0.01 threshold? and in addition why you didn't move miR-29a-3p or miR-30c-5p or miR-20a-5p to the logistic regression, they also have significant p-values?”

Response: We used 0.1 as cutoff for the multivariate linear regression analyses. Table 3 only shows the data for the significant variables, not for all variables included. We realized that was not clear in the original manuscript, therefore we included this explanation and Table S3 in “Suplements” with data from all variables in the linear regression.

Comment: “figure 1: I understood that the authors were building heatmap as they were building the logistic regression model, but it isn't the correct way to build a heatmap. I would recommend to build a heatmap with all p-value signifcant miRNAs and a second one with only the miRNAs selected in the final model.”

Response: We believe that the confusion addressed above regarding the variables significant in the linear regression contributed to this comment. The heatmaps were constructed for the three models resulting from the multivariate analyses, in other words, all the ones with significant p values. This was revised in the manuscript for clarity.

Reviewer 2 Report

Line 163: how the miRNAs were selected. The word 'discarded' meant these 18 miRNAs were not analyzed in the study?

Line 168: why these 2 miRNAs were used for normalization, while they are known to be associated with cancers.

Table 3: 3 miRNAs only were included in the model, while there are 19 miRNAs with p <0.1

Table 4: suggest adding combination of markers in the analysis as a risk score then calculate time-dependent ROC.

Suggest performing functional enrichment analysis for the final miRNAs and looking more in depth into gene targets within thyroid signaling KEGG pathway.

Author Response

We greatly appreciate the excellent review and valuable comments made on our work. After careful revision, we identified some methodological steps that were not presented as clearly as intended and may have been the reason for some of the comments.

All corrections were made on the document and are listed below. The changes are highlighted on the manuscript.

Once again, thank-you for your valuable comments and opportunity to review our work for your prestigious journal.

Comment: “Line 163: how the miRNAs were selected. The word 'discarded' meant these 18 miRNAs were not analyzed in the study?”

Response: The miRNAs were selected based on literature review. Considering our unique sample of metastatic patients and objective of finding miRNAs related to death we sought miRNA that were described as related to thyroid cancer but also with metastatic disease, tumor aggressiveness and epithelial mesenchymal transition, in thyroid cancer and others cancers. This was included in “Methods”. The results from specific miRNAs that did not meet quality standards for analyses (such as CT higher than 36 and detection in less than 70% of samples) were not included in the study (discarded). This was revised in the manuscript for clarity.

Comment: “Line 168: why these 2 miRNAs were used for normalization, while they are known to be associated with cancers”

Response: We used an endogenous miRNAs-based normalization strategy in which the microRNAs with the most stable expression are selected for normalization using the quantile method with the Expander software. Among our samples, these were let-7g-5p and miR-181a-5p.

The expressions of miR-181a-5p and miR-let-7g-5p in cancer are ambiguous and based on current literature, we cannot state that the miRs 181a-5p and let-7g-5p are oncogenes or tumor suppressors genes (discussion below). Also, we have to consider that all patients included in this specific study are metastatic, therefore this could explain why these microRNAs that are associated with cancer in some scenarios were stable in this study.

Wojciech Gierlikowski (2021) related that the expression of miR-181a-5p in the studied thyroid tissue pairs (normal adjacent tissue) was increased by 27% with a fold change of 2.27. The upregulation in tumor tissue vs normal tissue was also reported in thyroid cancer by Tian et al., 2022). In hepatocellular carcinoma (Ji et al, 2009), lung squamous cell carcinoma (Tian et al., 2016) the expressions was also increased, however, in breast cancer (Liu et al., 2020), non-small-cell lung cancer (Ma et al., 2015) and endometrial carcinoma (Yu et al., 2019) the miR-181a-5p expression was decreased in tumor tissues as compared to the adjacent tissues.

The same occurred with the miR-let-7g-5p expression that is reported as oncogene in colorectal cancer (Ozcan et al., 2016), and suppressor tumor in colorectal cancer (Niculae et al, 2022), in gallbladder carcinoma (Zhang et al, 2022), in epithelial ovarian cancer (Biamonte et al., 2019), in the serum of glioblastoma multiforme patients compared to normal controls (Dong et al, 2014), in nasopharyngeal carcinoma (Luan et al., 2016), in non-medullary thyroid carcinoma (Saiselet et al., 2016)and in thyroid cancer (Tian et al., 2022).

Comment: “Table 3: 3 miRNAs only were included in the model, while there are 19 miRNAs with p <0.1”

Response: All variables with p<0.1 were included in the multivariate linear regression analyses. Table 3 only shows the data for the significant variables, not for all variables included. We realized that was not clear in the original manuscript, therefore we included this explanation and Table S3 in “Suplements” with data from all variables in the linear regression.

Comment: “Table 4: suggest adding combination of markers in the analysis as a risk score then calculate time-dependent ROC.”

Response: This was done and Table 4 was modified to include analyses for the three proposed models.

Comment:” Suggest performing functional enrichment analysis for the final miRNAs and looking more in depth into gene targets within thyroid signaling KEGG pathway.”

Response: As suggested, we performed functional enrichment analysis using DAVID (https://david.ncifcrf.gov/) and String (https://string-db.org/). However, no specific  KEGG pathway related with thyroid was identified on first analyses. We appreciate the suggestion and will continue to explore this method.

Reviewer 3 Report

The manuscript by Leite et al., highlights specific miRNAs as prognosticators of death due to disease progression, in patients with metastatic papillary thyroid carcinoma. This is a very interesting finding. miRNAs have been shown to be diagnostic, prognostic, or predictive biomarkers in multiple cancer types, however to my knowledge, this is the first study to associate miRNAs with death in metastatic thyroid cancer. To this end, the present study is novel and of great scientific interest. Nevertheless, there are certain weaknesses that lower the significance of the findings:

Major issues:

1. The sample size for biomarker identification is quite small. The authors mention 108 patients in the beginning, but actually, only 17 plus 7 "ctrs" are included in the study. The authors should either magnify the sample, or alternatively, this should be clearly declared as a preliminary study.

2. There is no clear rationale why the specific miRNAs were chosen for inclusion in the present study. The authors mention that "were selected after extensive literature review to cover the different processes related to tumor behavior", but this statement needs further description.

3. Why were miRNAs let-7g-5p and miR-181a-5p used for normalization? Was there any exogenous control used, e.g. c-39 miRNA?

4. As the authors mention in the Discussion, miR-101 has been found to be a strong tumor suppressor in many types of cancer. However, according to the findings of the submitted study, miR-101 levels were found elevated in the deceased patients. The authors should provide other studies with relevant results, in order to explain this result.

Minor points:

1. The Materials and Methods section should be separated into subheadings to help the reader.

2. By Initial pT and pN classification, do you mean at diagnosis? Please clarify.

Author Response

We greatly appreciate the excellent review and valuable comments made on our work. After careful revision, we identified some methodological steps that were not presented as clearly as intended and may have been the reason for some of the comments.

All corrections were made on the document and are listed below. The changes are highlighted on the manuscript.

Once again, thank-you for your valuable comments and opportunity to review our work for your prestigious journal.

Comment: “The sample size for biomarker identification is quite small. The authors mention 108 patients in the beginning, but actually, only 17 plus 7 "ctrs" are included in the study. The authors should either magnify the sample, or alternatively, this should be clearly declared as a preliminary study”

Response: The sample is small and this is described as a limitation of the study. However, because of the indolent nature of PTC these cases are extremely rare, especially the ones resulting in death. This is the main reason why this outcome is so difficult to explore scientifically and there are so few studies on the subject. These were all the cases we could obtain primary tumor tissue sample and, to our knowledge, is the first study to do so for biomarker identification. This was better described in “Discussion” and the preliminary nature of the results was included in the Abstract and Discussion.

Comment: “There is no clear rationale why the specific miRNAs were chosen for inclusion in the present study. The authors mention that "were selected after extensive literature review to cover the different processes related to tumor behavior", but this statement needs further description.”

Response: The miRNAs were selected based on literature review. Considering our unique sample of metastatic patients and objective of finding miRNAs related to death we sought miRNAs that were described as related to thyroid cancer but also with metastatic disease, tumor aggressiveness and invasion processes such as epithelial mesenchymal transition, in thyroid cancer and others cancers. This was included in “Methods”.

Comment: “Why were miRNAs let-7g-5p and miR-181a-5p used for normalization? Was there any exogenous control used, e.g. c-39 miRNA?”

Response: We used an endogenous miRNAs-based normalization strategy in which the microRNAs with the most stable expression are selected for normalization using the quantile method with the Expander software. Among our samples, these were let-7g-5p and miR-181a-5p. We did not use exogenous controls. This was revised in the manuscript for clarity.

Comment: “As the authors mention in the Discussion, miR-101 has been found to be a strong tumor suppressor in many types of cancer. However, according to the findings of the submitted study, miR-101 levels were found elevated in the deceased patients. The authors should provide other studies with relevant results, in order to explain this result.”

Response: The literature shows that different microRNAs can have different roles in different tumors and specific situations and these processes are not yet fully understood.  The 6th and 7th paragraphs of Discussion were dedicated to this topic, and multiple references were cited (35-43) to discuss possible explanations for these results. The last paragraph of “Discussion” also addresses this topic.

Comment: “The Materials and Methods section should be separated into subheadings to help the reader”

Response: The template provided by Cancers does not state that this section can be divided in subheadings. However, for clarity, we rephrased the beginning some of the paragraphs.

Comment:  “By Initial pT and pN classification, do you mean at diagnosis? Please clarify.”

Response: Yes, at diagnosis. It was revised in the text for clarity.

Round 2

Reviewer 1 Report

The authors improved a lot the manuscript with more data and figures. But still there are some additions to the work to be made:

1. The authors explained about why choosing a p-value threshold to filter significant DE miRNAs for their regression model, it is recommended to change the text to 0.05 the cut-off, since all the chosen miRNAs are under this value and also it brings more reliability for the work, since no one uses a threshold of p-value 0.1, it is a high number, usually, articles uses 0.05 or under that value. 

2. It is nice that the authors compared their results with published microarray publications in the same area. However, it should be added a paragraph explaining the limitations of the model, since the authors are using a limited number of miRNAs to evaluate the expression, so it is unknown if there are other miRNAs that could contribute to the linear regression since they weren't evaluated. The authors are using a limited number of miRNA to start the experiment, so they can assure that some other miRNAs can also help this model.

Author Response

Dear reviewer, thank-you once again for your valuable review. The response to your comments are listed bellow and changes highlighted in the manuscript. 

Comment: "The authors explained about why choosing a p-value threshold to filter significant DE miRNAs for their regression model, it is recommended to change the text to 0.05 the cut-off, since all the chosen miRNAs are under this value and also it brings more reliability for the work, since no one uses a threshold of p-value 0.1, it is a high number, usually, articles uses 0.05 or under that value. "

Response: The threshold used for inclusion in the multivariate analysis was 0.1.  There are four variables with p between 0.05 and 0.1 that were also included in the model (miRs let-7i, 30a, 138 and 200b - table S3).  This method is employed to increase statistical rigor since it excludes possible dependent and confounding variables. The threshold for statistical significance was p<0.05. This was revised in the manuscript for clarity. 

Comment: "2. It is nice that the authors compared their results with published microarray publications in the same area. However, it should be added a paragraph explaining the limitations of the model, since the authors are using a limited number of miRNAs to evaluate the expression, so it is unknown if there are other miRNAs that could contribute to the linear regression since they weren't evaluated. The authors are using a limited number of miRNA to start the experiment, so they can assure that some other miRNAs can also help this model."

Response: This limitation was included in Discussion, as suggested. 

Reviewer 3 Report

The authors have successfully dealt with all of the proposed comments. I believe that the manuscript is now suitable for publication. However, I believe that the Methods section should be subdivided into paragraphs with clear subheadings to help the reader. In any case, this is a minor issue and does not hinder publication.

Author Response

Dear Reviewer, 

Thank-you very much for your valuable review. 

The Methods section was divided in the manuscript, as suggested. 

Kind regards,